# Transphyseal Hematogenous Osteomyelitis: An Epidemiological, Bacteriological, and Radiological Retrospective Cohort Analysis

**DOI:** 10.3390/microorganisms11040894

**Published:** 2023-03-30

**Authors:** Blaise Cochard, Céline Habre, Nastassia Pralong-Guanziroli, Nathaly Gavira, Giorgio Di Laura Frattura, Giacomo Di Marco, Christina N. Steiger, Geraldo De Coulon, Romain Dayer, Dimitri Ceroni

**Affiliations:** 1Pediatric Orthopedics Unit, Pediatric Surgery Service, Geneva University Hospitals, CH-1211 Geneva, Switzerland; 2Division of Orthopedics and Trauma Surgery, Geneva University Hospitals, CH-1211 Geneva, Switzerland; 3Pediatric Radiology Unit, Geneva Children’s Hospital, Geneva University Hospitals, CH-1211 Geneva, Switzerland

**Keywords:** growth plate, transphyseal osteomyelitis, *Kingella kingae*, MSSA

## Abstract

Transphyseal hematogenous osteomyelitis (THO) is a serious condition that can affect the growing physis, yet it is insufficiently recognized in children. The aim of this study was to explore the prevalence and epidemiology of pediatric THO, and to discuss the underlying pathophysiology. All consecutive cases of acute and subacute osteomyelitis admitted to our institution over 17 years were retrospectively studied. Medical records were examined for patient characteristics, bacteriological etiology, and medical and surgical management. Magnetic resonance imaging was reviewed for all patients to identify those with transphyseal spread of infection. For positive cases, the surface area of the transphyseal lesion was estimated relative to the total physeal cross-sectional area. Fifty-four (25.7%) of the 210 patients admitted for acute or subacute osteomyelitis were diagnosed with THO. The study population’s ages ranged from 1 month to 14 years old (median age 5.8 years, interquartile range 1–167 months). Fourteen (25.9%) patients were younger than 18 months old; the remaining 40 (74.1%) had a mean age of 8.5 years old. The most common sites of THO were the distal tibia (29.1%), the proximal tibia (16.4%), and the distal fibula (14.5%). Transphyseal lesions were due to acute infection in 41 cases and to subacute osteomyelitis in 14 cases. The two most frequently identified pathogens were *Staphylococcus aureus* (49.1%) and *Kingella kingae* (20.0%). An average transphyseal lesion represented 8.9% of the total physeal surface, and lesions comprised more than 7% of the physeal cross-sectional area in 51% of cases. Our study revealed that pediatric THO was more frequent than commonly thought. Transphyseal lesions were frequently above this 7% cut-off, which is of paramount importance since subsequent growth is more likely to be disturbed when more than 7% of the physeal cross-sectional area is injured. THO also affected children older than 18 months, an age at which transphyseal arterial blood supply to the epiphysis is believed to have disconnected. This finding suggests another pathophysiological reason for the transphyseal diffusion of infection, a topic deserving further studies and greater understanding.

## 1. Introduction

Transphyseal hematogenous osteomyelitis (THO) is an infectious process involving both the metaphysis and the epiphysis by extension of the infectious process across the physis in the immature skeleton. Depending on the virulence of the incriminated germ and the host’s response to infection, THO can have either an acute or subacute onset [1]. Subacute osteomyelitis is described as any osseous hematogenous infectious process lasting more than 2 weeks without acute symptomatology [2]. This specific condition is characterized by moderate localized bone pain, mild or no systemic manifestations, non-contributory laboratory results, negative blood cultures, but positive radiological findings [2,3,4,5,6,7,8,9,10,11,12].

Most osteoarticular infections (OAI) are primarily haematogenous in origin and result from asymptomatic or symptomatic bacteriemia. Among all origins, the respiratory tract is the most favorable portal to pathogens. From a general point of view, the most responsible pathogens for pediatric OAI are *Kingella kingae* and *Staphylococcus aureus*. *K. kingae* accounts for 50% of all osteoarticular infections and is mainly present in the 6–48 months old population [13]. *S. aureus* is responsible for 29% and is mainly found in the 5–15 year old population [14]. Nevertheless, atypical pathogens can be found in each age, hence the need to isolate and identify the responsible pathogen. From a pathophysiological perspective, because of long bones’ vascular characteristics, the main starting points for infection are currently considered to be their well-vascularized metaphyses. Indeed, their spongiosa are irrigated by abundant terminal blood vessels with leaky endothelia and sluggish flow that ends in capillary loops [15,16]. This vascular environment is thus conducive to the deposition of bacteria. In neonates and infants up to 18 months old, it is established that the presence of transphyseal vessels allows the bidirectional translocation of bacteria, from or to the epiphysis [16,17,18]. From 2 years old, arterial blood supply to the epiphyses becomes relatively disconnected from the metaphyses, and some authors have hypothesized that, beyond this age, hematogenous osteomyelitis could no longer cross the growth plate [15]. However, a recent paper identified an unexpectedly high prevalence of THO in children aged from 2 to 16 years old who had undergone magnetic resonance imaging (MRI) for pyogenic osteomyelitis, contrasting with this condition’s commonly taught pathophysiology [19].

THO constitutes a serious pediatric condition because a significant area of growing cartilage may be damaged, with potentially severe consequences for future bone development and function [20]. The present study’s objectives were thus to investigate the prevalence of THO in acute and subacute pediatric hematogenous osteomyelitis, to identify their epidemiological characteristics, and to explore their bacteriological etiology.

## 2. Materials and Methods

This retrospective study was approved by the Geneva Children’s Hospital Review Board (CE 14-102R). The medical charts of all consecutive pediatric patients aged from 1 day to 16 years old admitted with either acute or subacute hematogenous osteomyelitis from January 2006 to December 2022 were retrospectively reviewed. Our 111-bed tertiary pediatric hospital serves the city of Geneva and surrounding areas and is the only facility providing inpatient and specialized medical services for pediatric osteoarticular infection to the region’s 460,000 inhabitants.

Diagnosis codes for acute and subacute osteomyelitis were used to identify the population of interest from our institution’s electronic medical records. Children’s risks of having a bone infection were estimated using the criteria established by Morrey [21]. Inclusion criteria were confirmed cases of acute or subacute hematogenous infection discriminated based on the duration of symptoms before admission (i.e., less than 2 weeks versus from 2 weeks to 3 months, respectively), positive imaging studies (plain radiography or MRI), and the isolation of a pathogen in blood cultures or bone specimens. Only patients with complete radiological imaging of the anatomical region affected (plain radiography and MRI) were included in the study. Exclusion criteria comprised a diagnosis of chronic osteomyelitis (i.e., symptoms lasting more than 3 months), previous surgery, or a foreign body at the site of infection.

### 2.1. Epidemiological Investigations

Data collected included age, sex, symptom duration, signs and symptoms at presentation, and the bone involved. Initial laboratory studies were collected for each patient and included white blood cell count, platelet count, erythrocyte sedimentation rate, and C-reactive protein.

### 2.2. Microbiological Methods

Microbiological data included pathogen names, sources of positive cultures (blood or bone samples), and susceptibility to antibiotics, mainly methicillin and clindamycin. Blood cultures were systematically used to isolate the microorganisms responsible for hematogenous osteomyelitis. The present study’s blood culture media were BACTEC 9000 for the period before 2009 and an automated blood culture system (BD BACTEC FX) after that. Bone aspirate samples were sent to the laboratory for Gram staining, cell count, and immediate inoculation onto Columbia blood agar (incubated under anaerobic conditions), CDC anaerobe 5% sheep blood agar (incubated under anaerobic conditions), chocolate agar (incubated in a CO_2_-enriched atmosphere), and brain–heart broth. These media were incubated for 10 days. Two polymerase chain reaction (PCR) assays were also performed to identify bacteria when standard cultures were negative. Initial aliquots (soft tissue or fluid samples) (100–200 µL) were stored at −80 °C until processing for DNA extraction. A universal broad-range PCR amplification of the 16S rRNA gene was performed using BAK11w, BAK2, and BAK533r primers (Eurogentec, Seraing, Belgium). 

According to Cherkaoui et al. [22,23], the PCR broad-range process was conducted as follows. Firstly, samples were incubated at 55 °C for 1 h with proteinase K and lysis buffer. DNA was then extracted with a MagNAPure LC instrument using the MagNAPure LC DNA isolation kit II (Roche Molecular Biomedical, Mannheim, Germany). The final elution volume was 100 L. Five microliters of DNA extracts were used for each PCR analysis. Prior to PCR amplification, all employed reagents were mixed and incubated for 30 min at 37 °C with 10 U of Exonuclease III (Boehringer, Mannheim) in a 37 L master mix (10 mM Tris-HCl (pH 8.3), 50 mM KCl, 2.5 mM MgCl2, 0.75 U of Amplitaq LD (Perkin-Elmer, Cheshire, United Kingdom) and Limulus amoebocyte lysate reagent water (Cambrex LAL Kit). The enzyme was then inactivated by incubation at 70 °C for 20 min, following which 0.5 mM of each deoxynucleoside triphosphate, 30 pmol of each primer (Eurogentec, Seraing, Belgium) and 5 L of template DNA were added. The PCR amplification was started according to the followed process. The first amplification round was conducted with 30 pmol primers BAK11w and BAK2 with initial denaturation at 95 °C for 10 min and cycled as follow: 95 °C for 20 s, 48 °C for 45 s and 72 °C for 1 min for 40 cycles. Then, 5L of the first round of amplification was amplified in a semi-nested reaction with BAK11w and BAK533r. The second round of amplification was carried out in a similar manner, except that cycling was as follows: 95 °C for 20 s, 50 °C for 45 s, 72 °C for 1 min for 40 cycles. Amplicons were resolved on a 2% agarose EDTA gel, visualized by using ethidium bromide staining under UV illumination, using 174/HaeIII size markers (Promega). For phylogenetic identification, sequences were compared with sequences of known bacteria listed in official databases using the BLAST program available at the National Center for Biotechnology Information (http://www.ncbi.nlm.nih.gov/BLAST/ accessed on 12 January 2023). The criteria used to assign a species name were 98-100% homology to more than one GenBank sequence of the same species.

As of 2007, we also used a real-time PCR (rtPCR) assay targeting the *K. kingae* gene’s rtx toxin [23]. This assay is designed to detect two independent gene targets from the *K. kingae* rtx toxin locus, namely *rtxA* and *rtxB* [23]; it was used to analyze different biological samples, including synovial fluid, bone or discal biopsy specimens, and peripheral blood. In accordance with Cherkaoui et al. [23], real-time PCRs for *K. kingae* were performed using TaqMan Universal PCR Master Mix with AmpErase UNG (Applied Biosystems). This last was used with 0.5 μL of each primer (rtxA and rtxB), 0.25 μM of the probe, 5 μL input DNA and nuclease-free water (Promega). Duration of DNA extraction/purification was 1 h and rtPCR amplification and analysis was 3 h. Since September 2009, we have also carried out oropharyngeal swab PCR for children aged from 6 months to 4 years old. It has been demonstrated that this simple technique for detecting *K. kingae* rtx toxin genes in the oropharynx provides strong evidence that this microorganism is responsible for osteoarticular infection when positive and even stronger evidence that it is not responsible when negative [24]. The virulence factors of the pathogen were investigated in cases of extensive anatomical involvement. These virulence factors were the accessory gene regulator (Agr), Panton Valentine Leucocidin (PVL), and Toxin Shock Syndrome (TSS). Agr, which globally controls the production of virulence factor, was identified by rtPCR with specific primers [25]. PVL, which has cytolytic activity on polymorphonuclear neutrophils, is identified by rtPCR using luk-F and luk-S primers [26]. Toxin present in the TSS was identified by specific PCR as well. The methicilin sensitivity of *S. aureus* was determined by antibiogram and by specific rtPCR targeting mecA gene. Each of those rtPCR processes was based on the same procedure as with rtx toxin, but using different primer.

### 2.3. Radiological Investigations

In our institution, all children with suspected acute or subacute hematogenous osteomyelitis undergo an imaging work-up before beginning antibiotic therapy or a surgical procedure. In all patients, conventional radiographs were taken of the limb of concern. MRI was performed for all patients within 24 h of admission, with images acquired at 1.5-T (Avanto, Siemens, Erlangen, Germany) using the following sequences: tridimensional short tau inversion recovery (STIR), T1-weighted turbo spin-echo (one longitudinal plane); two orthogonal planes with T2-weighting plus fat suppression, STIR (longitudinal plane), and a water-only dataset of fast spin-echo T2-weighted Dixon sequences (axial plane); diffusion-weighted imaging (axial plane) using single-shot echo planar imaging, acquired using two *b* values, 0 and 800 s/mm^2^, with an automatic, mono-exponential calculation of the apparent diffusion coefficient maps; and post-contrast injection T1-weighted spin echo with frequency-selective fat saturation (two orthogonal planes). Post-contrast sequences were obtained after injecting 0.2 mL/kg of gadoteric acid (Dotarem, Guerbet, France).

Both radiographs and MRI were retrospectively reviewed by a board-certified pediatric radiologist (C.H.; 5 years of experience with pediatric MRI) and a senior pediatric orthopedist (D.C.; 30 years of experience with pediatric osteomyelitis) to find a consensus diagnosis. Pediatric MRI scans were analyzed to determine the location of transphyseal infection and the ratio between the lesion and the total surface of the physis involved. Surface area estimations were obtained using manual segmentation of the transphyseal lesion and the physis from their 3-D reconstruction and the mean intensity projection of the tridimensional STIR sequence along the plane of the physis (OsiriX MD v. 12.0.1 software, Geneva, Switzerland).

### 2.4. Statistical Analysis

Descriptive analyses were used for patients’ characteristics, for bacteriological results and for surface calculation of transphyseal injury on MRI. The prevalence of THO was calculated relative to the total of patients admitted for acute and subacute hematogenous osteomyelitis on the same observational period. All statistical analyses were performed using Jamovi (The Jamovi project (2022). Jamovi (Version 2.3). Retrieved from https://www.jamovi.org accessed on 12 January 2023).

## 3. Results

The medical files of 210 children who had been treated for acute or subacute osteomyelitis were listed during the period studied. Among this cohort, 72.9% (153/210) presented with long bone involvement and 27.1% (57/210) presented with carpal or tarsal bone involvement. Fifty-four patients (29 boys and 25 girls) were revealed to have suffered THO (25.7%). Their mean age at admission was 78.9 months old ± 60.4 months (range 1–167 months). Fourteen patients (25.9%) were less than 18 months old, and 18 (33.3%) were aged between 6 and 48 months. One patient presented with multifocal osteomyelitis involving the left distal femur and proximal tibia, and bacteriological samples were taken from both of these points. Thus, our epidemiological and biological data were based on 54 patients, whereas our bacteriological and radiological data were based on 55 cases. From an etiological viewpoint, 74.5% (41/55) of cases of transphyseal involvement were secondary to acute osteomyelitis, whereas, in 25.5% (14/55) of cases, it was due to subacute osteomyelitis.

THO was located on an upper limb in 12.7% (7/55) of cases, with 87.3% (48/55) of cases involving a lower limb. The distal tibia was the most frequently affected location (16/55), followed by the proximal tibia (9/55) and the distal fibula (8/55). The other bones involved are listed in Table 1.

Nineteen (35.2%) children were febrile at admission, with a mean temperature of 37.6 °C; 32 (59.3%) children had had a documented prodromal illness involving a temperature higher than 38 °C during the weeks before their hospital admission. Mean white blood cell count was 10,800 ± 4100/mm^3^ (range 4100–210,000/mm^3^) and was considered elevated in 20/54 patients, with a band shift expressed in six of them. A C-reactive protein level was available for all the patients and was considered elevated (>10 mg/L) in 38 (70.4%) of them and normal in the remaining 16. The mean C-reactive protein level across all patients was 50.8 ± 50.5 mg/L (range 0.3–200 mg/L). The erythrocyte sedimentation rate was measured in 36 patients and was abnormal (>20 mm/h) in 23 (63.9%) of them. The mean erythrocyte sedimentation rate was 35.2 ± 28.5 mm/h (range 3–115 mm/h). The mean platelet count was 344,000 ± 134,000/mm^3^ and was abnormal (>400,000/mm^3^) in 28.8% of patients.

Identification of a microorganism was possible in 47 patients (85.5%) through blood (cultures or PCR) and/or in operative samples (cultures or PCR of bone aspirate). Pathogens were recovered from blood cultures in 11 of the 41 examinations performed (26.8%). In 15 additional cases, PCR assays were performed after negative blood cultures, and microorganisms were identified in six of these cultures (40%). Pathogen identification in bone specimens was possible in 20 of the 33 standard isolation experiments performed (60.6%) and in two of the 9 PCR assays performed (22.2%). In 21 cases (38.9%), no bone aspirate was performed. The most commonly identified causative pathogens of osteoarticular infection in our study were methicillin-sensitive *Staphylococcus aureus* (MSSA) (27 cases; 49.1%), *Kingella kingae* (11 cases; 20.0%), and *Streptococcus pneumoniae* (two cases; 3.6%). In eight cases (14.5%), no pathogens were identified despite the adequate bacteriological investigations carried out. Finally, microorganisms were found in seven other cases (Table 2). Among the MSSA-infected population, two patients produced PV leucocidin and four produced the toxin for toxic shock syndrome. In addition, four were carriers of Agr type 1, one carried type 2, and four carried type 3. When only considering the results for the 47 bacteriologically confirmed cases of THO, *K. kingae* was responsible for 23.4% of them, and MSSA was responsible for 57.4%. Despite an adequate bacteriological investigation, no pathogens were identified in 14.5% of cases.

All the children underwent a conventional radiography of their affected limb segment at admission, and 25.5% (14/55) of the images were suggestive of transphyseal osteomyelitis. MRI contributed to this diagnosis for every patient. Transphyseal lesions represented, on average, 8.9% of the physis surface, with a range from 1% to 31.2%. Lesions were less than 7% of the total physeal area in 25 children, from 7–15% in 16, from 15–25% in 8, and were more than 25% in two (Table 3). In four cases, the surface area of the physeal damage could not be analyzed due to insufficient imaging quality. These were two patients with a *Streptococcus pneumoniae* infection, one with *Salmonella*, and one with MSSA. Transphyseal lesions were more extensive in cases of osteomyelitis due to *S. aureus* than those due to *K. kingae.*

## 4. Discussion

For a long time, clinicians considered that initial bacterial invasions specifically affected the metaphysis since this part of the bone has a particular local vascular anatomy and little phagocytic macrophage activity. It was also thought that because transphyseal vessels did not persist after a certain age the growth cartilage constituted an effective barrier to metaphyseal hematogenous acute osteomyelitis diffusing towards the epiphysis. However, over time, we have come to understand that this situation is not so straightforward, and some concepts which seemed immutable require re-evaluation. The advent of MRI and its extensive use for documenting osteoarticular infections has shown, for example, that acute osteomyelitis could start on the epiphysis [27] and that it may display transphyseal diffusion far more frequently than was suspected [19]. Thus, the present study aims to determine the prevalence of THO in acute and subacute pediatric hematogenous osteomyelitis, to identify their epidemiological characteristics, and to explore their bacteriological etiology. This will allow a better understanding of this pathology, which is still under-researched, and to improve its management.

The present work represents the largest consecutive case series examining the prevalence of transphyseal damage during osteomyelitis, whether acute or subacute. Our results showed that 25.7% of cases of pediatric hematogenous osteomyelitis crossed the growth plate. Even if this rate is substantial and significant, we remain far from the 81% frequency of pyogenic transphyseal osteomyelitis evoked by Gilbertson-Dahdal et al. [19]. The cases of THO reported in their study may have been due to the fact that its authors were specifically studying infections due to pyogenic pathogens, which are known to cause localized suppuration and bone tissue necrosis. We could therefore assume that pyogenic germs have a greater destructive effect on growth cartilage. Furthermore, the prevalence of THO rises in our study to 34.4% when only considering long bone involvement.

Interestingly, our study also found that only 25.9% of children with transphyseal osteomyelitis were younger than 18 months old. This observation is interesting since it is commonly accepted that only infants under 18 months present with a more permeable growth plate due to the communicating metaphyseal and epiphyseal vessels that enable the transphyseal diffusion of a metaphyseal infection. Two anatomical studies formed the basis of this theory. Chung studied arterial supply to the proximal end of the femur in 150 specimens from autopsied fetuses and children and suggested that the epiphyseal plate constituted an absolute barrier to blood flow between the epiphysis and metaphysis in the great majority of them [28]. Trueta examined the femoral head’s normal vascular anatomy during growth and noted that the ascending metaphyseal arteries were rapidly decreasing in number and caliber at 18 months of age. He demonstrated that after 18 months, the metaphyseal arteries entered the epiphysis by crossing the growth plate’s outer perimeter but without piercing the growing cartilage. Trueta also noted that the metaphyseal vessels diminished still further in number and caliber over the following years, eventually probably disappearing completely after the fourth year [15]. Since Trueta’s initial work, several authors have questioned this view, pointing out that growing cartilage only provided a relative barrier [18,29] to spreading infections.

Thus, the presence of a vascular connection between the metaphysis and epiphysis could constitute a plausible explanation for transphyseal osteomyelitis in children younger than 18 months old or, at least, younger than 4 years old. However, in older children, the diffusion of the infection across the growing cartilage would undoubtedly have to involve either the unexpected persistence of transphyseal vessels or a different lesional process. As mentioned, several authors harbored doubts about growing cartilage’s impermeability to infection and, thus, its role as a barrier to spreading infection. Another pathophysiological hypothesis involves the bacteria lodged in the junction between the physis and the metaphysis—thus adjoining the physeal cartilage—causing an infectious erosion of the growing cartilage. A direct attack on the growing cartilage can be quick and brutal when pyogenic pathogens are involved (e.g., MSSA producing PV leucocidin; Figure 1). On the contrary, infectious erosion may be slower when due to less aggressive microorganisms (such as *K. kingae*; Figure 2). This last point suggests that the present study’s over-representation of subacute osteomyelitis in THO may validate this explanation.

Our results suggest that not all bones are equally susceptible to THO. Indeed, leg bones are more prone to present such a lesion, especially the tibia, which accounts for nearly half of cases alone. This predominance of the involvement of leg bones is probably not the result of chance. Legs are probably the body segments most frequently subjected to blunt trauma. It is unanimously recognized that closed direct trauma may play a significant role in the genesis of osteomyelitis. Thus, by pushing this reasoning to the extreme, one could hypothesize that repetitive local trauma might play a role in the genesis of THO; we could easily assume that blunt trauma is responsible for bone hyperemia, facilitating both osteomyelitis and the transphyseal spread of infection.

The mean surface area of transphyseal lesions represented 8.9% of the physis surface; that area was significantly greater than 7% in 26/55 cases. This 7% cut-off is of paramount importance since the size of the physeal bridge is recognized as the main factor affecting future bone growth. In animal models, it has been demonstrated that subsequent growth was more likely disturbed when more than 7% of the physeal cross-sectional area was injured [30]. For bars involving less than 7% of the physeal cross-sectional area, there are currently strong arguments seeming to show that growth of the surrounding uninjured areas of the physis can cause traction and ultimately fracture the bar, allowing for a resumption of normal growth [31]. From a bacteriological viewpoint, our results suggested that transphyseal involvement is more extensive during infections due to MSSA than due to *K. kingae*. Future studies should involve follow-up to reveal whether transphyseal involvement and its bacterial etiology play significant roles in residual growth.

Another point emerging from our study was that two pathogens were responsible for most of the cases of transphyseal osteomyelitis, i.e., MSSA and *K. kingae*. When only considering the results of the 47 cases confirmed bacteriologically, these two pathogens were detected in 80.8% of cases, and *K. kingae* was only found in children under 4 years old. *K. kingae*’s high prevalence among cases of THO is a crucial factor since this germ is known to be responsible for subacute osteomyelitis, which frequently spreads through the physis.

This unexpectedly high incidence of THO in children aged between 2 and 16 years old also emphasizes the importance of performing MRI in any case of osteomyelitis, whether acute or subacute, since only this examination can confirm whether an infection has spread through the physis. Moreover, MRI is the examination of choice for quantifying the surface area of the growing cartilage damaged by the infection. In this respect, the present study demonstrated that damage to the growing cartilage through infection may be focal or, on the contrary, very extensive. Thus, recognizing and quantifying a transphyseal lesion is crucial because this may have prognostic value about the proper functioning of the physeal cartilage, which will require strict monitoring of its future growth.

Our study had some limitations. Its retrospective nature increased the risk of missing certain cases because of medical coding errors. This also increased the proportion of missing data and patients lost to follow-up. Nevertheless, the descriptive material examined provided a lot of information about rates of transphyseal lesions due to osteomyelitis. These results should be confirmed and enriched with a future real multi-center study, which would allow us to examine a larger number of patients and thus determine the risk that transphyseal osteomyelitis may pose to subsequent bone growth. Another relevant study limitation was that quantifying the surface area of the growing cartilage damaged by the infection should be considered as an estimation as physes are not planar, and thus may not represent the full extent of the transphyseal infection.

## 5. Conclusions

The present study confirmed that 25.7% of cases of acute or subacute pediatric hematogenous osteomyelitis may be due to an infection crossing the growth plate. However, this ratio rises to 34.4% if we consider only long bone involvement. This observation goes beyond the concept that young infants show greater growth plate permeability due to their communicating metaphyseal and epiphyseal vessels and the involvement of other pathophysiological processes. In more than 50% of cases, transphyseal lesions cover more than 7% of the surface of the affected physis. This finding is of paramount importance since, beyond this cut-off, the size of a physeal bridge is recognized as the main factor affecting future bone growth. Thus, the results of this work deserve further examination to define what may be the future in terms of residual growth of such THO.

## Figures and Tables

**Figure 1 microorganisms-11-00894-f001:**
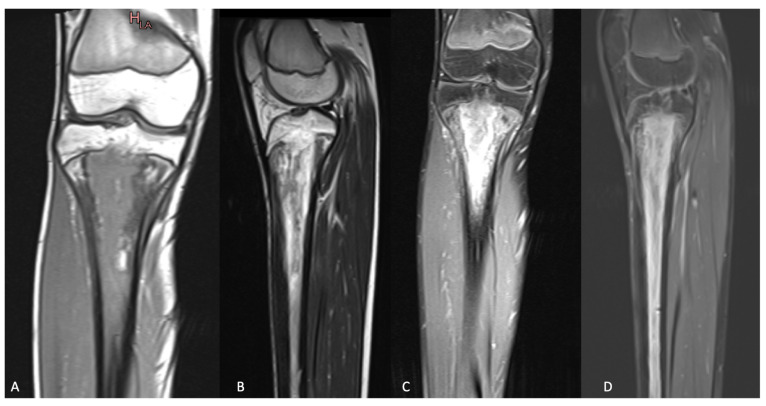
MRI of the right knee showing primary metaphyseal acute hematogenous osteomyelitis with secondary pandiaphyseal and transphyseal extension due to methicillin sensitive S. aureus, producer of the Panton-Valentin Leucocidin: (**A**) coronal view of the right knee with T1 sequence; (**B**) sagittal view of the right knee with T2 sequence; (**C**) coronal view of the right knee with T1 sequence and Gadolinium; (**D**) sagittal view of the right knee with T1 sequence and Gadolinium.

**Figure 2 microorganisms-11-00894-f002:**
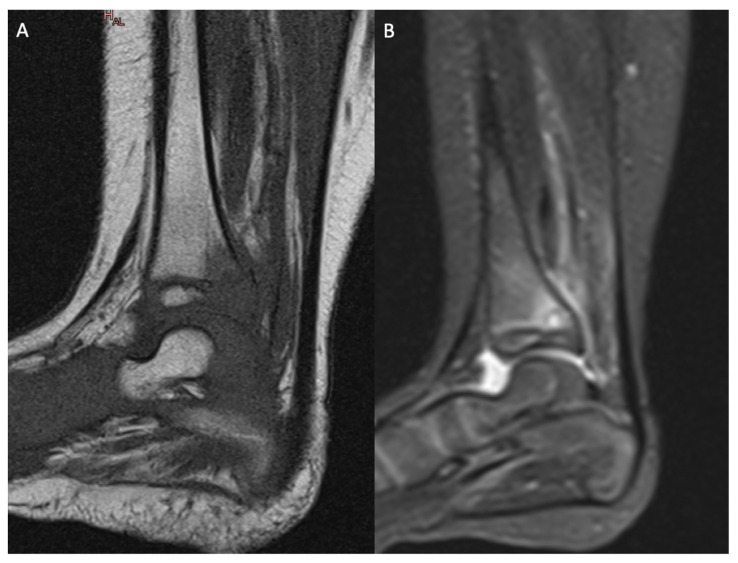
MRI of the left ankle showing primary metaphyseal acute hematogenous osteomyelitis with transphyseal extension to the distal tibia due to *Kingella kingae*: (**A**) sagittal view of the left ankle with T1 sequence; (**B**) sagittal view of the left ankle with T2 STIR sequence.

**Table 1 microorganisms-11-00894-t001:** Distribution of bone segment involvement.

Frequencies of Localization			
Localization	Counts	% of Total	Cumulative
Distal tibia	16	29.1%	29.1%
Proximal tibia	9	16.4%	45.5%
Distal fibula	8	14.5%	60.0%
Proximal femur	7	12.7%	72.7%
Distal femur	6	10.9%	83.6%
Proximal humerus	3	5.5%	89.1%
Proximal fibula	2	3.6%	92.7%
Distal humerus	2	3.6%	96.4%
Distal radius	1	1.8%	98.2%
Distal ulna	1	1.8%	100.0%

**Table 2 microorganisms-11-00894-t002:** Pathogen frequencies.

Frequencies of Pathogen			
Pathogen	Counts	% of Total	Cumulative
Methicillin-Sensitive Staphylococcus Aureus	27	49.1%	49.1%
Kingella Kingae	11	20.0%	69.1%
Unknown	8	14.5%	83.6%
Streptococcus pneumoniae	2	3.6%	87.3%
Streptococcus Pyogenes Gr. A	2	3.6%	90.9%
Salmonella (serovar Typhimurium sp)	2	3.6%	94.5%
Polybacterial (MSSA, Staphylococcus epidermidis, Streptococcus sp)	1	1.8%	96.4%
Staphylococcus epidermidis	1	1.8%	98.2%
Escherichia Coli	1	1.8%	100.0%

**Table 3 microorganisms-11-00894-t003:** Distribution of physeal involvement relative to the total surface area of growing cartilage (I: <7%; II: 7–15%; III: 15–25%; IV: >25%).

Frequencies of Distribution of Physeal Involvement			
Distribution of Physeal Involvement	Counts	% of Total	Cumulative %
I	25	49.0%	49.0%
II	16	31.4%	80.4%
III	8	15.7%	96.1%
IV	2	3.9%	100.0%

## Data Availability

Data are available on reasonable request.

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
