# Peer review of "Transphyseal Hematogenous Osteomyelitis: An Epidemiological, Bacteriological, and Radiological Retrospective Cohort Analysis"

_microorganisms, 2023, doi:10.3390/microorganisms11040894_

Round 1
Reviewer 1 Report
In this retrospective cohort study, the authors analyzed epidemiology, bacteriology, and radiology characteristics of transphyseal hematogenous osteomyelitis (THO) in pediatric patients in an institution for over 17 years. This study is designed well and conveys new and important information regarding THO, which can help clinicians better understand this disorder. Several issues should be addressed or revised before acceptance.
1. Title: “Transphyseal Hematogenous Osteomyelitis” would be more precise.
2. Results: Are there differences regarding the characteristics of epidemiology, bacteriology, and radiology issues between patients with THO and those without THO? If possible, I would recommend that the authors try to analyze potential risk factors to develop THO.
3. Results: How about the clinical efficacy of the patients with THO? Did the clinical efficacy of patients with THO differ from those without THO?
Author Response
We should like to thank the reviewer very much for his/her interest and his/her thoughtful comments on our paper.
- In accordance with his/ her recommendation, the title of the article has been modified as such: “Transphyseal Hematogenous Osteomyelitis: An epidemiological, bacteriological, and radiological retrospective cohort analysis. »
- Are there differences regarding the characteristics of epidemiology, bacteriology, and radiology issues between patients with THO and those without THO?
- No there is no difference
If possible, I would recommend that the authors try to analyze potential risk factors to develop THO.
-
- We have tried to analyze our results in search of any risk factor to develop THO but without success. The retrospective nature of the study does not allow such an approach.
- How about the clinical efficacy of the patients with THO and did the clinical efficacy of patients with THO differ from those without THO?
- To our knowledge, this parameter has never been evaluated. It is therefore incumbent upon us to study the repercussion of this type of THO on subsequent growth. However, it is to be feared that children who have presented THO involvement may present more complications, particularly with regard to growth. We will not fail in the future to carry out a study of this type with a long-term follow-up.
Reviewer 2 Report
Dear authors,
In this cohort analysis you sought to investigate the prevalence of THO in pediatric hematogenous osteomyelitis.
Please find my comments below.
With regard to the introduction section, I would advise you to organize it more appropriately from a structural point of view. In particular I would recommend you divide this section into three paragraphs of equal length.
For the materials and methods, I would suggest you provide the IRB number at the beginning of this section.
Also, ‘pidemiological’ should be ‘epidemiological’.
Table 2. MSSA needs to be spelled out.
Discussion section.
The first paragraph needs to be revised for clarity. Also, an opening paragraph should include different information than the comparison you’ve made between past and present clinical practice. In particular, the rationale behind the research question should be mentioned in the first paragraph and then the main findings of the paper at the end of the opening paragraph.
Again, I believe that discussion section should be better organized and utilization of subtitles could be useful to prevent confusion of the readers.
Author Response
- Concerning the comment: “With regard to the introduction section, I would advise you to organize it more appropriately from a structural point of view. In particular I would recommend you divide this section into three paragraphs of equal length.”
- In accordance with his/her recommendation, the structure of the introduction was modified accordingly to his/her comment.
- Concerning the comment: “For the material and methods, I would suggest you provide the IRB number at the beginning of this section”
- In accordance with his/her recommendation, the IRB number was added at the beginning of this section.
- Concerning the comment: “‘pidemiological’ should be ‘epidemiological’.”
- After a detailed review of our article, we were unable to find this spelling error.
- Concerning the comment about the discussion: “The first paragraph needs to be revised for clarity. Also, an opening paragraph should include different information than the comparison you’ve made between past and present clinical practice. In particular, the rationale behind the research question should be mentioned in the first paragraph and then the main findings of the paper at the end of the opening paragraph.”
- The discussion section has been re-organized as requested by the reviewer.
- The different themes were addressed in clearly separated paragraphs.
- Concerning the comment about the discussion: “Again, I believe that discussion section should be better organized and utilization of subtitles could be useful to prevent confusion of the readers.”
- The use of subtitles does not in any way correspond to the publication requirements defined by the journal in question
Round 2
Reviewer 1 Report
The authors have addressed all of my concerns appropriately, and thus, may be accepted for publication.
Author Response
Dear reviewer,
We would like to thank you for your work and your recommendations for the improvement of our manuscript.